# Transcriptome Analysis of the Necrotrophic Pathogen *Alternaria brassicae* Reveals Insights into Its Pathogenesis in *Brassica juncea*

Sivasubramanian Rajarammohan[a]

aAgricultural Biotechnology Division, National Agri-Food Biotechnology Institute, Mohali, Punjab, India

**ABSTRACT**   *Alternaria* blight or leaf spot caused by *Alternaria brassicae* has an enormous economic impact on the *Brassica* crops grown worldwide. Although the genome of *A. brassicae* has been sequenced, little is known about the genes that play a role during the infection of the host species. In this study, the transcriptome expression profile of *A. brassicae* during growth and infection was determined. Differential expression analysis revealed that 4,430 genes were differentially expressed during infection. Weighted gene coexpression network analysis helped identify 10 modules, which were highly correlated with growth and infection. Subsequent gene ontology (GO) enrichment analysis of the modules highlighted the involvement of biological processes such as toxin metabolism, ribosome biogenesis, polysaccharide catabolism, copper ion transport, and vesicular trafficking during infection. Additionally, 200 carbohydrate-active enzymes (CAZymes) and 80 potential effectors were significantly upregulated during infection. Furthermore, 18 secondary metabolite gene clusters were also differentially expressed during infection. The clusters responsible for the production of destruxin B, brassicicene C, and HC-toxin were significantly upregulated during infection. Collectively, these results provide an overview of the critical pathways underlying the pathogenesis of *A. brassicae* and highlight the distinct gene networks that are temporally regulated. The study thus provides novel insights into the transcriptional plasticity of a necrotrophic pathogen during infection of its host. Additionally, the *in planta* expression evidence for many potential effectors provides a theoretical basis for further investigations into the effector biology of necrotrophic pathogens such as *A. brassicae*.

**IMPORTANCE**   *Alternaria brassicae* is a necrotrophic pathogen that can infect almost all members of the *Brassicaceae* family. *A. brassicae* causes extensive yield losses in oilseed mustard and has practically restricted the cultivation of oilseed brassicas in regions with cool and foggy climatic conditions (foothills and mountainous terrains) where the severity of the pathogen is the highest. In this study, I identified the differentially expressed genes associated with the pathogenicity of *A. brassicae* through transcriptome sequencing. Also, I have been able to delineate pathways that are active during the early and late stages of infection. Consequently, this study has provided crucial insights into the molecular mechanisms underlying the pathogenesis of *A. brassicae*, an important necrotrophic pathogen.

**KEYWORDS**   *Alternaria brassicae*, transcriptome, effectors, necrotrophs, *Brassica*, CAZymes

Address correspondence to siva.r24@nabi.res.in.

The author declares no conflict of interest.

Alternaria blight or leaf spot is endemic to the *Brassica* crops grown in the tropical and subtropical regions worldwide. It is caused by a wide variety of pathogenic species from the ubiquitous fungal genus *Alternaria* (1). Among the *Alternaria* spp. invading *Brassicaceae*, *A. brassicae*, *A. brassicicola*, *A. alternata*, and *A. raphani* reportedly cause the most damage (2). *A. brassicae* is a necrotrophic pathogen that affects its host species at all stages of growth and symptoms of infection. *A. brassicae* is one of the dominant invasive

species on oleiferous brassicas. *Brassica juncea* is widely grown in the Indian subcontinent, Australia, and parts of Europe as an oilseed crop. *A. brassicae* is a major constraint to realizing the yield potential of *B. juncea* in the Indian subcontinent. Understanding the molecular mechanisms employed by the pathogen during the infection is essential for identifying novel targets for disease management. The pathogenicity of the related species, *A. alternata*, has been attributed to secondary metabolites or host-specific toxins (3). Although several toxins, effectors, and virulence factors of *A. brassicicola* have been characterized, the mechanism by which it colonizes the host remains unknown (4). Similarly, the mechanisms/pathways underlying the pathogenesis of *A. brassicae* are also unknown. Studies on the molecular aspects of *A. brassicae* pathogenicity have primarily concentrated on the roles of secondary metabolite toxins such as destruxin B (5, 6). The study of other pathogenicity factors in *A. brassicae* was precluded by the hypothesis that destruxin B was a host-specific toxin and that detoxification of the toxin could lead to the abolition of infection (7). Recent studies confirmed that destruxin B is not a host-specific toxin, and other proteinaceous toxins/effectors may be responsible for the pathogenicity of *A. brassicae* (8).

High-quality genomes of various *Alternaria* species are currently available with the advent of newer and less expensive sequencing technologies (9–12). While this has accelerated the study of pathogen biology and the identification of various effectors and pathogenicity factors, functional evidence for most identified genes in the form of transcriptomic data is lacking. Transcriptome analysis of *Alternaria* leaf spot infections has primarily focused on the plant responses to infection (13–16). Additionally, these transcriptome studies were performed with cDNA microarrays; hence, retrieving fungal reads/transcripts from the raw data is not feasible.

Given the enormous economic impact of *A. brassicae* worldwide and the lack of a comprehensive transcriptomic profile of *A. brassicae* during infection, I undertook the current study to (i) determine the transcriptome expression profile of *A. brassicae* during growth and infection, (ii) identify and functionally annotate the differentially expressed genes (DEGs) during different infection stages, (iii) identify and classify gene coexpression networks that are differentially regulated temporally, and (iv) analyze the repertoire of carbohydrate-active enzymes (CAZymes), secondary metabolite-encoding gene clusters, and effectors that are differentially expressed *in planta* during infection.

## RESULTS

**Sequencing and assembly.** *A. brassicae* is a slow-growing pathogen compared to other phytopathogens of the *Alternaria* genus. Therefore, 7-day-old mycelium from potato dextrose agar (PDA) plates was collected to obtain enough biomass for RNA extraction. Two and 4 days post inoculation (dpi) were chosen as the early and late time points of infection since previous studies have shown that *A. brassicae* spores maximally penetrate the host tissue at 2 dpi and visible symptoms are clearly visible at 4 dpi (17). The maximum number of droplets was inoculated on the leaf surface to obtain an optimal yield of *in planta* fungal mycelia (Fig. 1B). RNA from both plate-grown fungi (*in vitro*) and infected leaf tissue (*in planta*) was sequenced by paired-end sequencing, yielding approximately 218 million reads (Fig. 1A; see also Table S1 in the supplemental material). Approximately 92.8% of the reads from the *in vitro* samples mapped to the *A. brassicae* reference genome. In contrast, 0.2 to 1.6% of the 2-dpi *in planta* samples and 1.3 to 3.1% of the 4-dpi *in planta* samples mapped to the *A. brassicae* genome (Table S1). The lower percentage of mapping in *in planta* samples has also been observed in other pathogen-host interaction studies (18–21). In light of the new transcriptome data set generated for *A. brassicae*, the genome was reannotated to identify any novel transcripts or genes that were missed earlier. A total of 231 novel protein-coding transcripts were identified. These transcripts were then merged with the older annotations to generate a newer set of annotations (V 2.0), and this version was used for all further analyses. A gene was defined to be expressed in a particular sample if >10 reads from a sample type could be mapped to the gene. A total of 11,290 (95.5%) genes were found to have detectable expression levels in at least one of the sample types (*in vitro* or *in planta*), of which 9,367 genes (83% of the expressed genes) were expressed in the *in planta*

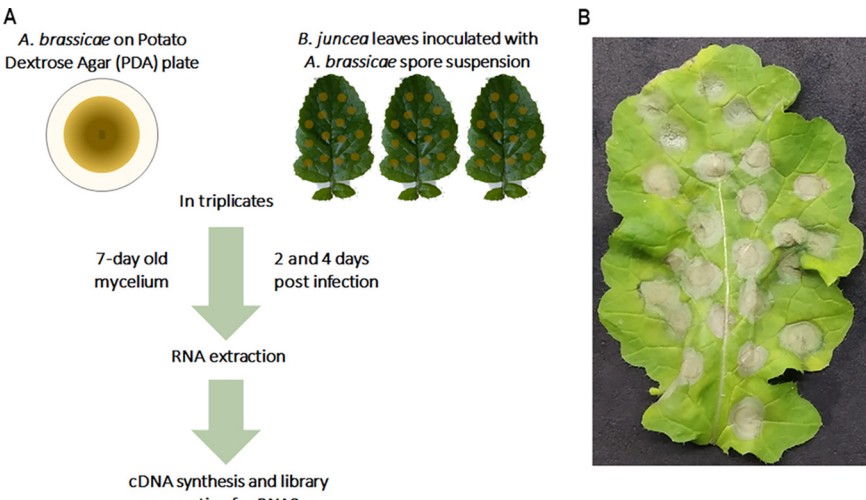

**FIG 1** An overview of the transcriptome analysis experimental design. (A) Schematic representation of the sample collection and RNA sequencing strategy for the transcriptome analysis of *in vitro* and *in planta* samples. (B) Representative image of *B. juncea* leaf demonstrating symptoms of *A. brassicae* infection (4 days postinfection [4 dpi]).

samples. Despite the low number of fungal reads in the *in planta* samples, they mapped to >80% of the total expressed genes of *A. brassicae*. Principal-component analysis (PCA) suggested that differences in gene expression could be mostly attributed to the sample type (*in vitro* versus *in planta*; PC1, 66%) and time of sampling (PC2, 20%) (Fig. 2). Furthermore, the biological replicates of each sample type clustered together, indicating high correlation within the samples of each type (*in vitro*, 2 dpi, and 4 dpi).

**Differential gene expression analysis of *A. brassicae* transcriptome.** Differential gene expression analysis was performed to identify *A. brassicae* transcriptome changes during the pathogenesis of *B. juncea*. A total of 2,600 and 3,495 *A. brassicae* genes were significantly differentially expressed at the early (2-dpi) and late (4-dpi) stages of infection,

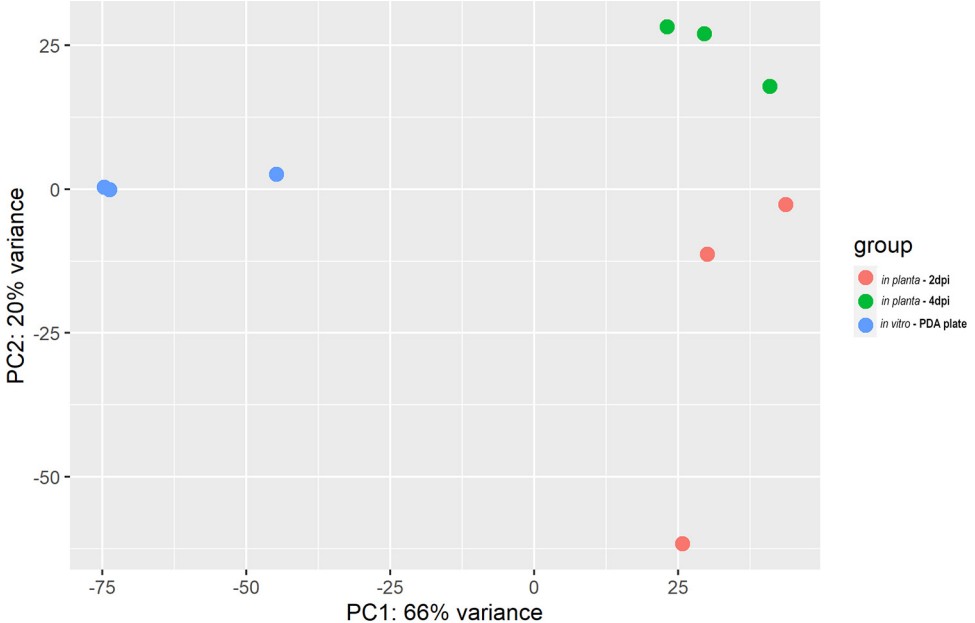

**FIG 2** Principal-component analysis. The PCA plot depicts that the samples in the transcriptome expression data are clustered distinctly by sample type (*in vitro* versus *in planta*) as well as the time of infection (early, 2 dpi, and late, 4 dpi).

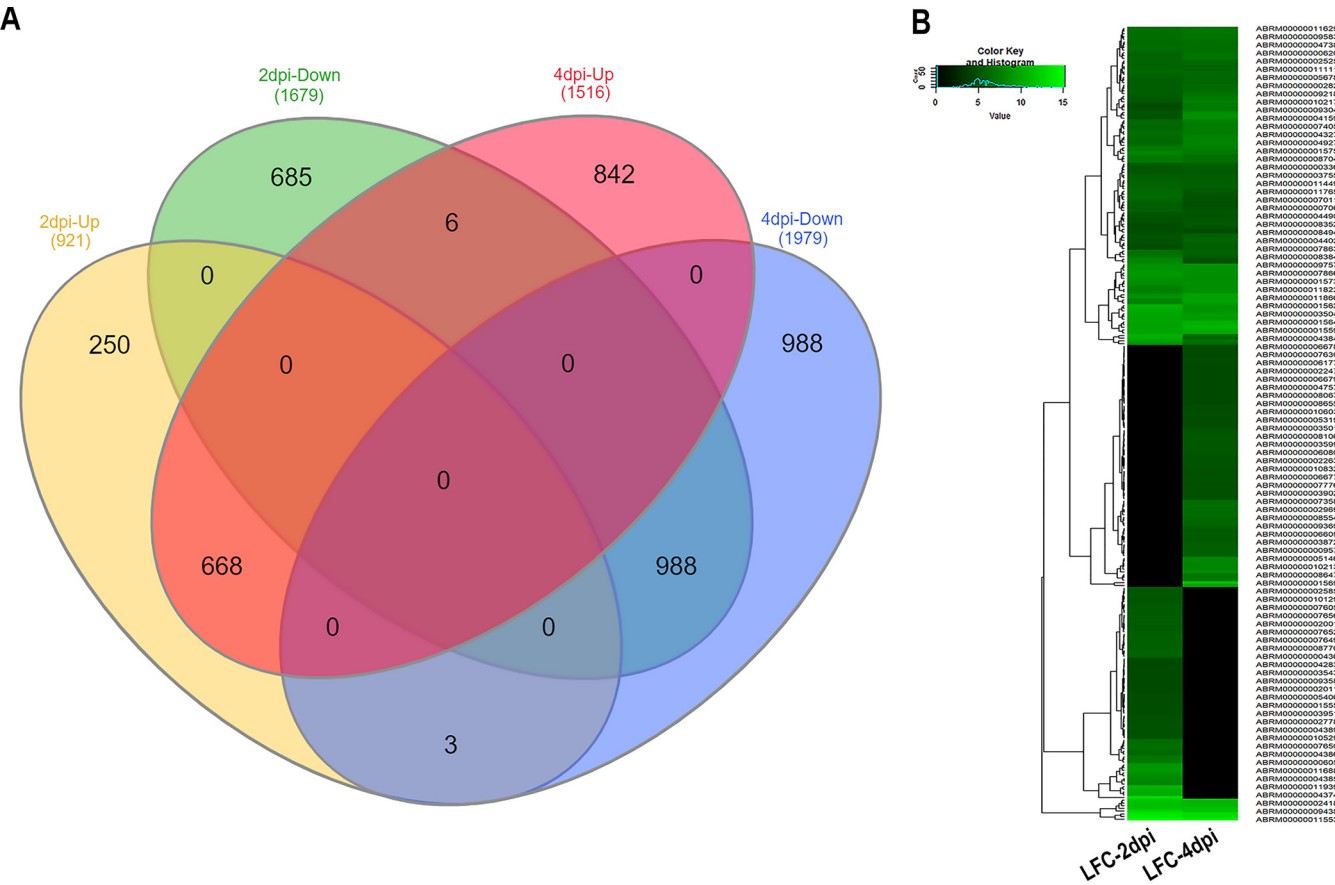

**FIG 3** Differentially expressed genes (DEGs) of *A. brassicae*. (A) Venn diagram representing the number of common and unique genes between up- and downregulated gene sets of 2 dpi and 4 dpi. (B) Heatmap depicting the expression profiles of the top upregulated genes [$\log_2(FC) \geq 4.64$; $FC \geq 25$] at 2 and 4 dpi.

respectively. Among these, 921 and 1,679 genes were up- and downregulated, respectively, at 2 dpi. Similarly, 1,516 and 1,979 genes were up- and downregulated, respectively, at 4 dpi (Fig. 3A; Table S2). A total of 203 and 214 genes were upregulated by >25-fold ($\log_2$ fold change |LFC|, >4.6) at 2 and 4 dpi, respectively. Among these highly upregulated genes, 127 were consistently upregulated by >25-fold at 2 and 4 dpi (Fig. 3B; Table S3). Furthermore, I also identified 250 and 842 genes that were specifically upregulated at 2 dpi and 4 dpi, respectively. A total of 668 and 988 genes were commonly up- and downregulated at 2 and 4 dpi, respectively (Fig. 3A). Three genes were significantly upregulated at 2 dpi but were downregulated at 4 dpi (Table S2), whereas six genes were significantly downregulated at 2 dpi but were upregulated at 4 dpi (Table S2).

**Functional annotation of DEGs.** Gene functional enrichment analysis was performed by assigning gene ontology (GO) terms to DEGs. DEGs were grouped according to their putative roles in biological process (BP), molecular function (MF), and cellular component (CC) per the GO consortium. GO enrichment analysis of DEGs at 2 and 4 dpi provided insights into the biological pathways that were temporally regulated during pathogenesis.

At the early infection stage, GO BP terms such as ribosome biogenesis, ribosome assembly, carbohydrate metabolism, pectin catabolism, toxin biosynthesis, and RNA processing were enriched in the differentially upregulated genes. GO BP terms such as regulation of chromosome organization, heterochromatin formation, cellular protein modification, autophagy, and amino acid transport were enriched in the downregulated genes at the early infection stage (Fig. 4; Table S4). Similarly, at the late infection stage, GO BP terms such as cell wall polysaccharide catabolism, cell wall organization, xylan/pectin catabolism, and RNA surveillance were enriched in the differentially upregulated genes. GO BP terms such as regulation of DNA replication, protein phosphorylation, regulation of transcription,

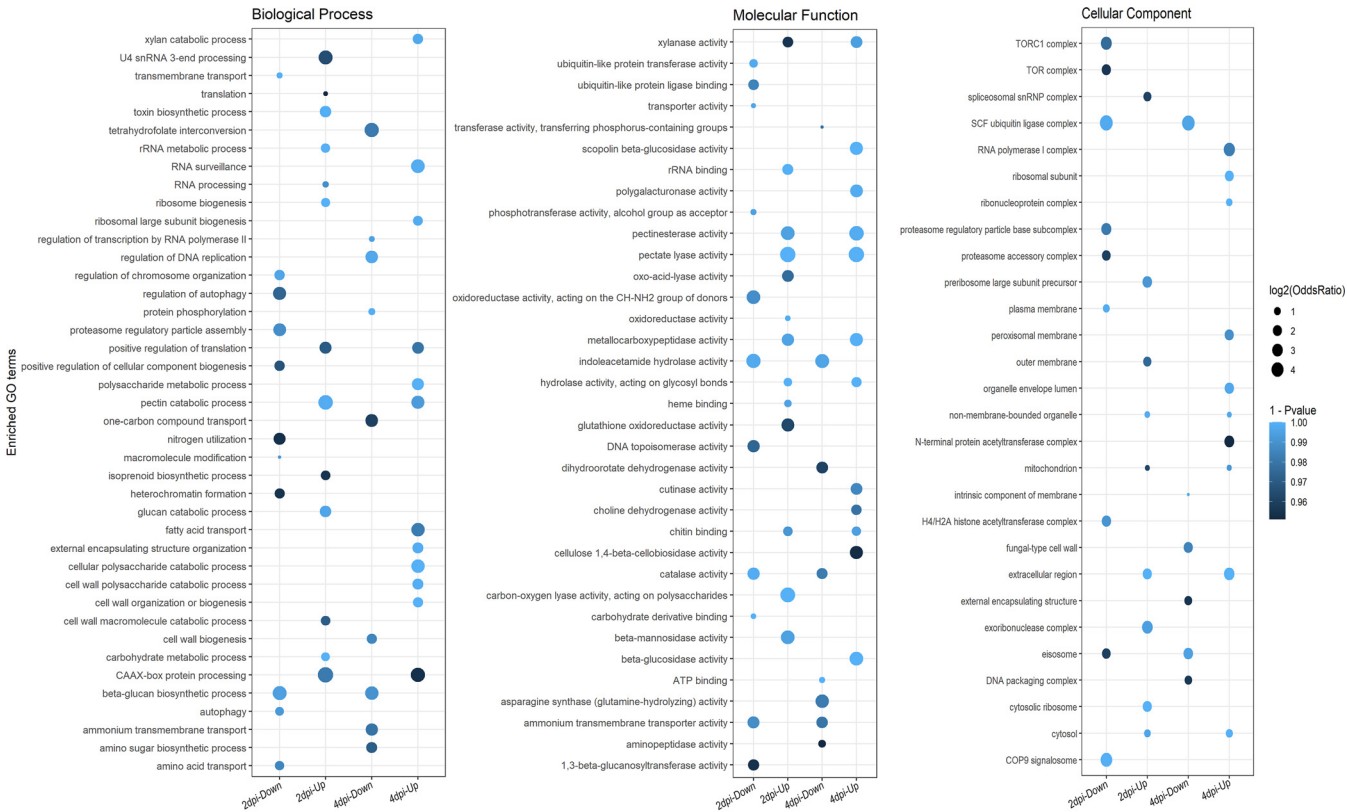

**FIG 4** GO enrichment analysis of DEGs. Dot plots depicting the GO enrichment of DEGs under biological process, molecular function, and cellular component categories. The color of the dots represents the *P* value of the hypergeometric test for GO enrichment, and the size of the dot represents the odds ratio of the GO term with respect to that in the background gene set.

amino acid metabolism, and ammonium transport were enriched in the differentially downregulated genes at the late infection stage (Fig. 4). Additionally, the molecular functions associated with the above processes were also similarly enriched in the respective gene sets (Fig. 4; Table S5).

**WGCNA reveals temporally regulated distinct networks.** To identify functional pathways involved in the pathogenesis of *A. brassicae*, highly coexpressed gene modules were inferred from the DEGs using weighted gene coexpression network analysis (WGCNA). A total of 4,430 DEGs were used to construct the coexpression network. In WGCNA, modules are defined as clusters of highly interconnected genes possibly having a functional relationship. Using a soft power threshold of 16 ($R^2 > 0.9$) and a minimum module size of 50, 13 modules with module sizes ranging from 72 to 2,015 genes were identified (Fig. S1, Fig. S2, and Table S6). The gene modules were correlated with the sample traits, i.e., growth on the PDA plate, early infection (2 dpi), and late infection (4 dpi). It was observed that 10 out of 13 modules were strongly correlated with at least one of the traits (Table S7). The turquoise (M1), dark green (M2), cyan (M3), and dark red (M5) modules were positively correlated with the growth on PDA plates, whereas the royal blue (M6) module was negatively correlated with the growth on PDA (Fig. S3). The yellow (M7) module was positively correlated with the early infection stage, whereas the green (M9) module was positively correlated with the late infection stage (Fig. S3). The dark turquoise (M4) module was positively correlated with growth on PDA but was negatively correlated with the early infection stage. Additionally, the light cyan (M8) and dark gray (M10) modules were negatively correlated with the early and late infection stages, respectively.

GO enrichment analysis of the genes in the M1, M2, M3, M4, and M5 modules revealed a significant overrepresentation of GO BP terms such as cell morphogenesis, primary and secondary metabolism, autophagy, regulation of DNA replication, tetrahydrofolate interconversion, amino acid biosynthesis and transport, protein phosphorylation, endosome

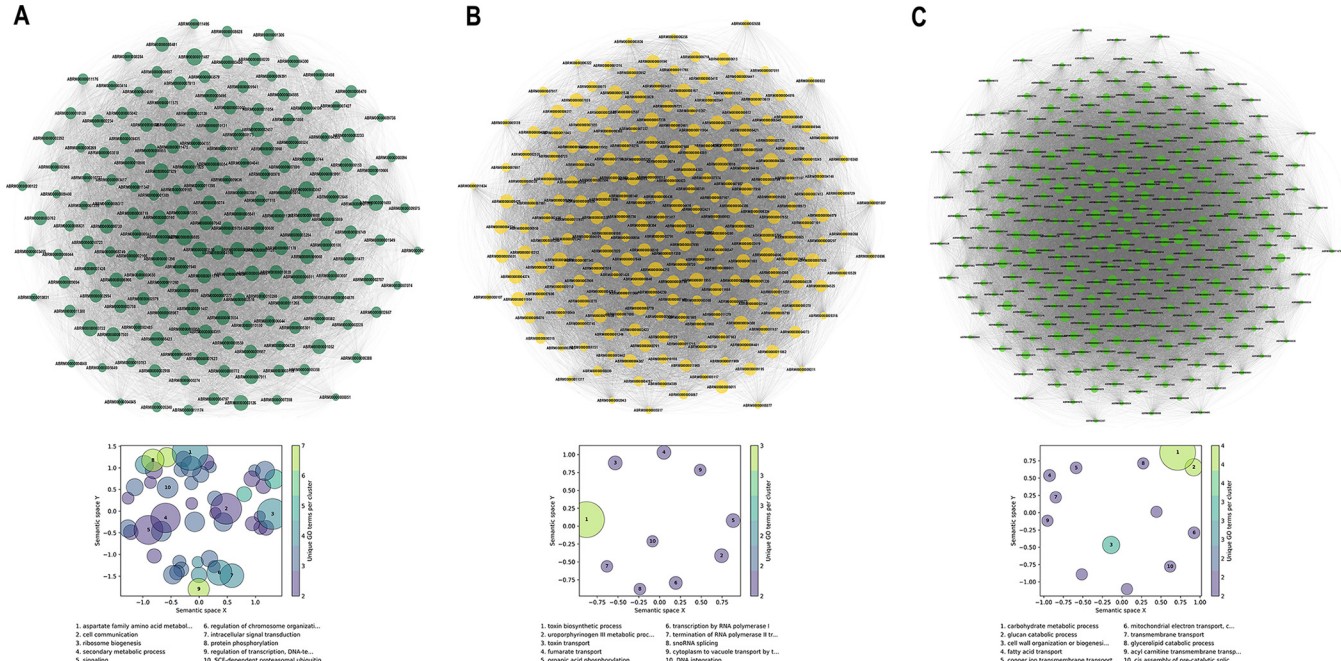

**FIG 5** WGCNA and GO enrichment analysis of DEGs. (A) Weighted gene coexpression network of the genes in the dark green (M2) module and GO terms enriched in the modules correlated with growth on PDA. (B) Weighted gene coexpression network of the genes in the yellow (M7) module and GO terms enriched in the modules correlated with the early infection stage (2 dpi). (C) Weighted gene coexpression network of the genes in the green (M9) module and GO terms enriched in the modules correlated with the late infection stage (4 dpi).

organization, cytosolic transport, regulation of RNA metabolism, transcription, and cell communication (Fig. 5A; Table S8). The overrepresentation of these pathway genes indicates that the transcriptional machinery is geared toward cell division and primary metabolism during the growth on the PDA. In contrast, in the M7 module, GO BP terms such as toxin biosynthesis process, secondary metabolite biosynthesis, toxin transport, transcription, and splicing were overrepresented (Fig. 5B; Table S8). Furthermore, the M9 module contained a significant overrepresentation of GO BP terms such as carbohydrate metabolic process, xylan/pectin catabolic process, vesicular trafficking, copper ion transmembrane transport, and cell wall organization (Fig. 5C; Table S8).

The network analysis followed by the GO enrichment analysis exhibits a clear demarcation in the pathways enriched at the three different stages studied. The growth phase on PDA was enriched for pathways that governed cell growth, division, and primary metabolism. Distinct pathways were enriched at 2 and 4 dpi, wherein the early stage (2 dpi) was enriched for pathways involved in RNA processing, ribosomal translation, and toxin metabolism. In contrast, the pathways at the late stage were predominantly enriched for degradation of the host cell wall components and vesicular trafficking.

**Candidate genes involved in the pathogenesis of *A. brassicae*. (i) CAZymes.** Most necrotrophic pathogens secrete various carbohydrate-active enzymes (CAZymes) to degrade and break down the plant cell wall. The CAZyme profile of *A. brassicae* was identified and reported previously (22). I cataloged the CAZymes that were significantly upregulated during the pathogenesis of *A. brassicae*. A total of 108 and 192 CAZymes were found to be significantly upregulated at 2 and 4 dpi, respectively. However, only 82 and 131 CAZymes significantly upregulated at 2 and 4 dpi, respectively, were secreted outside the cell. A total of 200 unique CAZymes were found to be significantly upregulated during the course of infection. The 200 CAZymes included 103 glycoside hydrolases (GHs), 12 glycosyl transferases (GTs), 25 carbohydrate esterases (CEs), 3 carbohydrate-binding modules, 39 auxiliary activity enzymes, and 18 polysaccharide lyases (PLs). Polysaccharide lyases (PLs) cleave uronic acid-containing polysaccharides via a $\beta$-elimination mechanism and are responsible for the degradation of pectin, one of the most abundant components of the plant cell wall. I identified 15 secretory PLs that were highly upregulated at both 2 and 4 dpi (Table S9). However,

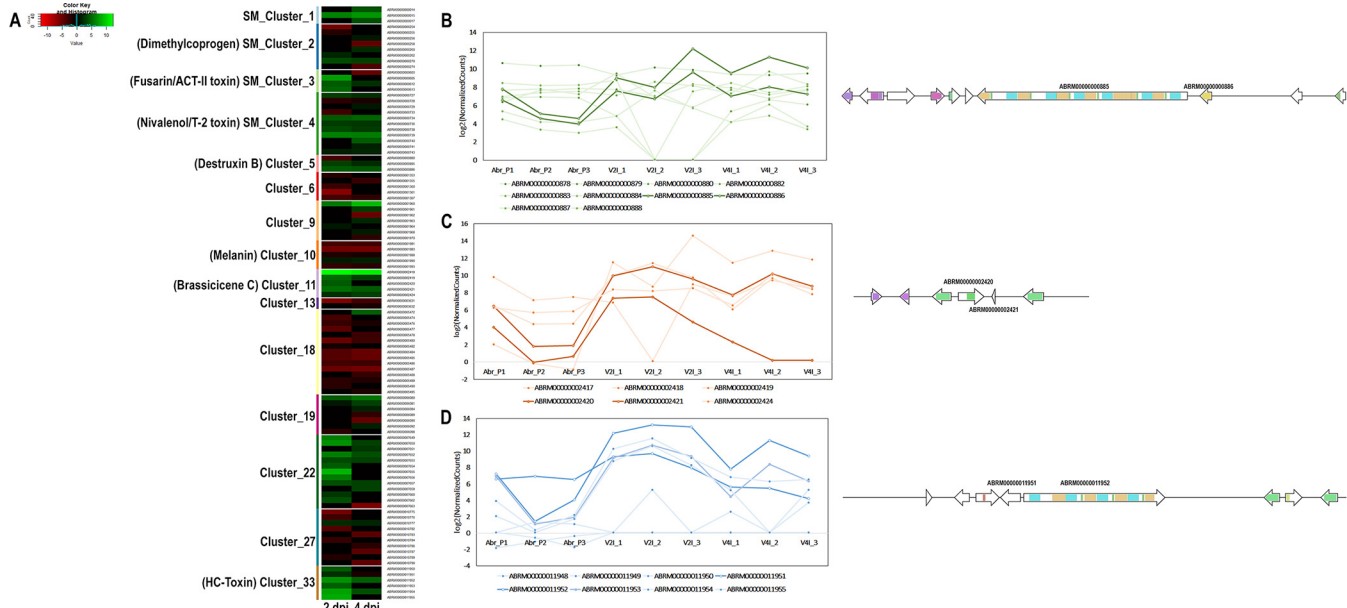

**FIG 6** Expression profiles of secondary metabolite gene clusters in *A. brassicae*. (A) Heatmap indicating the expression profiles of significantly up- or downregulated gene clusters. (B to D) Normalized expression values of the SM cluster genes encoding destruxin B (B), brassicicene C (C), and HC-toxin (D) in the nine samples. The lines denoting the core biosynthetic genes in panels B, C, and D are darkened.

it was observed that polygalacturonases (GH28 family), which are responsible for hydrolyzing the $\alpha$-1,4-glycosidic bonds in pectin, are exclusively upregulated during the late stage of infection (4 dpi) (Table S9).

**(ii) Effectors.** Effectors are small secretory proteins with distinct physicochemical properties that play an important role in establishing pathogenesis. Computational prediction of effectors in the *A. brassicae* genome had revealed 198 putative candidates (22). I further examined the expression status of these putative effectors during infection. It was observed that 80 of the 198 effectors were significantly upregulated during infection (Table S10; Fig. S4). Consistent expression across both the time points was observed for 49 of the 80 effectors expressed during infection. A total of 53 and 76 effectors were significantly upregulated at 2 and 4 dpi, respectively. Additionally, it was also observed that 29 of the 80 effectors were also characterized as CAZymes, thus indicating their pivotal role in the pathogenesis of *A. brassicae*. The AA9 family or the copper-dependent lytic polysaccharide monooxygenases (LPMOs) were reportedly expanded in the *Alternaria* genus (22). I identified six of the 11 previously reported secreted LPMOs in *A. brassicae* to be significantly upregulated at 4 dpi. However, only one of the LPMOs were significantly upregulated at 2 dpi. Other known effectors such as the conserved fungal extracellular membrane (CFEM) domain-containing proteins, Nudix effectors, and necrosis- and ethylene-inducing peptide 1-like proteins (NLPs) were found to be significantly upregulated at 2 and 4 dpi (Table S10). It was observed that both the previously characterized *A. brassicae* NLPs (AbrNLPs) (23) were significantly upregulated at 2 and 4 dpi.

**(iii) Toxins/secondary metabolites.** Secondary metabolites or nonproteinaceous toxins are known pathogenicity factors of many phytopathogens, including *Alternaria alternata*. Toxins are encoded by biosynthetic gene clusters that are usually coregulated. The *A. brassicae* genome was predicted to contain 34 secondary metabolite (SM) gene clusters (22). I evaluated the expression profile of the 34 SM clusters during growth and infection. A total of 18 SM gene clusters were differentially expressed during infection, among which 10 SM gene clusters were significantly upregulated and eight SM clusters were significantly downregulated. The melanin gene cluster was significantly downregulated during infection (Fig. 6; Table S11). In contrast, the siderophore (dimethylcoprogen), destruxin B, brassicicene C, and HC-toxin gene clusters were significantly upregulated during infection (Fig. 6; Table S11). Additionally, several other SM gene clusters involved in the

synthesis of unknown metabolites/toxins were also significantly upregulated during infection.

**(iv) Peptidases.** In fungi, peptidases are known to play an important role in several developmental processes, cellular signaling, and response to stress (24, 25). Some of the secreted peptidases in fungal pathogens are known to act as effectors/pathogenicity factors through modification of host protein machinery leading to suppression of host responses (26–28). The *A. brassicae* genome was found to contain 251 peptidases and peptidase-like proteins (22). However, only 66 of the 251 peptidases were found to be secreted outside the cell. A total of 114 peptidases were found to be differentially expressed during the course of infection (Table S12). Specifically, 31 and 49 peptidases were upregulated during 2 and 4 dpi, respectively. Additionally, a majority of the secreted peptidases (40 of 66) were differentially expressed during infection, indicating their prominent role during pathogenesis.

## DISCUSSION

In the present study, I investigated the alterations in the transcriptome profiles of *A. brassicae* growth on an artificial medium and during the infection of *B. juncea*, its natural host. Transcriptome studies on host-pathogen interactions are limited by the challenges in obtaining an adequate representation of pathogen sequences for further analyses (18–21). Therefore, I employed an inoculation method to maximize the enrichment of fungal transcripts in the infected leaf samples. Although the approach yielded fewer reads mapping to the reference genome, >80% coverage (9,367 of the 11,290 genes) of the transcripts could be obtained in the *in planta* samples. In total, 2,600 and 3,495 *A. brassicae* DEGs were identified at the early and late stages of infection, respectively. Functional annotation of the DEGs followed by GO enrichment analysis helped identify transcriptionally active key pathways/processes during different stages of infection.

The gene coexpression analysis and subsequent GO enrichment analysis have led to the identification of the infection program modulation across the two infection stages. During the course of *A. brassicae* infection, penetration through the stomata is initiated as early as 6 h postinoculation; however, maximum penetration sites are observed at 2 dpi (17). At 2 dpi, water-soaked lesions start to appear at the macroscopic level, whereas by 4 dpi clear necrotic lesions are visible on the leaf surface (17). Distinct changes in the sets of biological processes activated could be observed during growth and the two infection stages (Fig. 7). The colonization of *A. brassicae* caused marked changes in the transcriptional activity at 2 and 4 dpi. This is further supported by the observation that toxin metabolic processes, RNA processing, and ribosome biogenesis were enriched GO terms at 2 dpi (Fig. 7). The secondary metabolites and toxins such as destruxin B and brassicicene C are known to modulate plant physiology to facilitate penetration of the host and subsequently modulate the immune responses to the pathogen (6, 29). In parallel, RNA processing and ribosome biogenesis processes are required for the production of proteinaceous effectors and CAZymes that target host proteins and carbohydrates to cause necrosis and establish infection. At 4 dpi, the major GO terms that were enriched belonged to carbohydrate metabolic process, cell wall polysaccharide catabolism, and copper ion transport (Fig. 7). This indicates the temporal shift in the infection process from facilitation of colonization to the establishment of infection by cellular degradation of the host cells. The gene coexpression networks thus highlighted the temporal processes unique to each infection stage. This infection process is in line with the hypothesis that most necrotrophs do not immediately cause cell death upon interaction with the host and there may exist a short biotrophic phase before the onset of necrosis (30).

I observed that the pathogenicity factors in *A. brassicae* mostly consist of secreted CAZymes, toxins, and effectors. This is in contrast to *Alternaria alternata* pathovars, which use only specialized secondary metabolites or toxins as pathogenicity determinants (31–33). These results suggest that the necrotrophic lifestyle within the *Alternaria* genus itself has evolved through different genetic mechanisms. The plant cell wall is the primary barrier

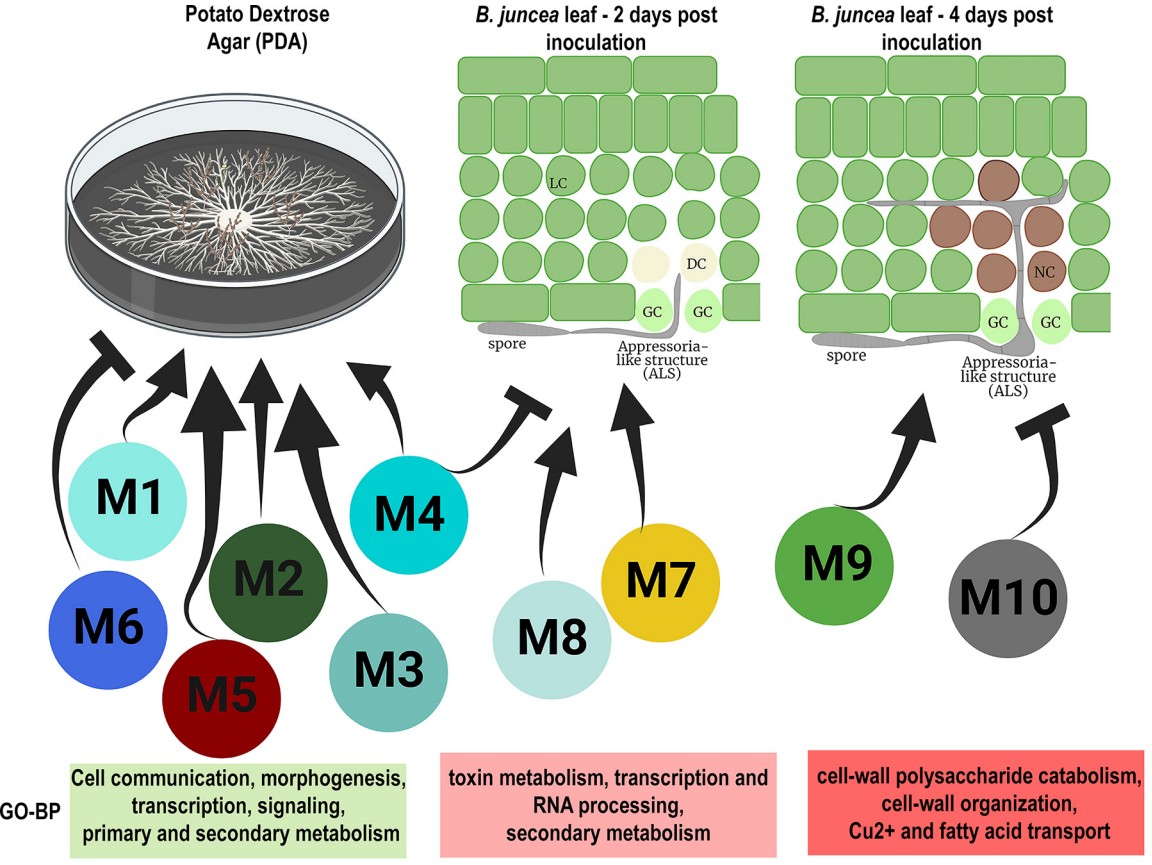

**FIG 7** Overview of the pathogenesis of *A. brassicae* in *B. juncea*. The different WGCNA modules positively/negatively correlated with the three stages of growth and infection are shown. GC, guard cells; DC, dying cells; NC, necrotic cells; LC, live cells. Module colors are represented per their assignment in WGCNA. Representative GO biological process terms of the modules are also listed.

encountered by microbial pathogens, and the complexity of the plant cell wall polysaccharides determines basal resistance to most invading microorganisms. CAZymes comprise several classes of enzymes that metabolize different carbohydrate moieties (34). They are often classified as cell wall-degrading enzymes (CWDEs) in the context of phytopathogens since they process or hydrolyze plant cell wall polysaccharides to facilitate infection and acquire nutrients from the hydrolyzed polysaccharides (35). The diversity of the CAZymes secreted by the pathogen, in turn, affects its host range. It was observed that 36% of the total CAZymes in the *A. brassicae* genome were significantly upregulated during infection. A markedly higher number of secreted CAZymes (see Table S10 in the supplemental material) that are also classified as effectors point to their role as one of the main determinants of pathogenicity in *A. brassicae*. The PL and CE families were the most active enzyme families during the infection process, wherein 81% and 64% of the total PLs and CEs were expressed, respectively. Pectin is a major component in the cell wall of dicot plants and the middle lamellae that help bind cells together (36). The digestion of pectin causes tissue collapse, tissue necrosis, and subsequent release of oligopectin compounds that may trigger plant defense (37). Pectin-digesting enzymes such as pectate lyases (PLs) and pectinesterases that digest pectin are, therefore, considered crucial pathogenicity factors. Loss-of-function mutations of PLs in many phytopathogens have contrasting effects on pathogenicity. In the closely related *A. brassicicola*, loss of function of two pectate lyases reportedly has opposing effects on the pathogenicity of the strain (38, 39). This indicates that functional redundancy exists in the case of some PLs, as observed in the case of other phytopathogens such as *Cochliobolus carbonum* (40). The PL family in *A. brassicae* consists of 21 members, among which 17 were significantly upregulated during infection. This indicates that the PLs in *A. brassicae* may be acting in coordination or may be functionally redundant.

A systematic study of loss-of-function mutations of these PLs would be required to delineate their individual functions and their redundancy.

Secreted effectors are critical players in mediating pathogenesis. Putative effector candidates were identified in *A. brassicae* through computational approaches (22). I determined the expression profiles of the putative effector candidates to evaluate their role during the infection process. Approximately 59% of the putative effectors were expressed at least at one of the time points. Similar transcriptome studies in other necrotrophs such as *Sclerotinia sclerotiorum* and *Botrytis cinerea* also observed >50% of the putative effectors to be differentially expressed during infection (18, 20, 41). This indicates that proteinaceous effectors are vital during the pathogenesis of *A. brassicae* and necrotrophic pathogens in general. This is in contrast to the earlier view that secondary metabolites or toxins are major players in pathogenicity in *A. brassicae* (5–7). Although a large proportion of the *in planta*-expressed effectors do not have any known function or domain, I identified a few effectors that have been characterized previously in other phytopathogens. However, apart from AbrNLPs, no other effector from *A. brassicae* has been characterized. This study, therefore, provides *in planta* expression evidence for a majority of the putative effector candidates and lays the foundation for further functional characterization of *A. brassicae* effectors.

Pathogens belonging to the *Alternaria* and *Cochliobolus* genera are well-known producers of secondary metabolites and peptide toxins that enable them to infect their hosts. The results reveal that some of the SM biosynthetic clusters in *A. brassicae* are transcriptionally active during infection of *B. juncea*. Some SM clusters code for known secondary metabolites, whereas most transcriptionally active clusters code for unknown or unidentified secondary metabolites. Destruxin B, a cyclic depsipeptide and a key pathogenicity factor of *A. brassicae*, is reportedly a host-specific toxin of *A. brassicae* (5). Destruxin B has not been reported to be produced by any other *Alternaria* species. The results suggest that the core biosynthetic genes of the destruxin B cluster, i.e., *DtxS1* and *DtxS3*, are significantly upregulated during infection. Destruxin B reportedly contributes to the aggressiveness of the pathogen but does not act as a host-specific toxin (8). The SM cluster coding for brassicicene C was significantly upregulated during infection. Brassicicene C belongs to the fusicoccane family, which are diterpenoids with various effects on plant physiology, including the opening of stomata (29). Therefore, destruxin B and brassicicene C may aid in pathogenesis by modulating the plant physiology but may not be directly involved in the pathogenesis. HC-toxin is the major virulence determinant of *Cochliobolus carbonum*, and resistance against *C. carbonum* is conferred by a carbonyl reductase gene that detoxifies HC-toxin (42). Two species of the *Alternaria* genus, *viz.*, *A. jesenskae* and *A. brassicae*, are the only other fungal species that contain the SM cluster that produces this toxin (22, 43). The HC-toxin cluster is also one of the highly upregulated gene clusters during the infection of *B. juncea*. The presence of the HC-toxin cluster in *A. brassicae* may also be one of the factors contributing to its wider host range than that of the other pathogens in the *Alternaria* genus.

In conclusion, the transcriptome of *A. brassicae* infection has revealed the mechanisms by which the pathogen rewires the cellular processes during growth and infection. The analysis also highlights a temporal regulation of distinct gene networks that exists to facilitate and establish pathogenesis. Additionally, the study has provided *in planta* expression evidence for many potential effector candidates and will thus facilitate further research into the effector biology of necrotrophic pathogens such as *A. brassicae*. Consequently, this will enable the development of novel strategies for integrated disease management in crop plants.

## MATERIALS AND METHODS

**Plant material and fungal strain.** The fungal strain *A. brassicae* J3 (44) was used for the infection assays in the expression analysis. The strain was grown on potato dextrose agar (PDA; pH adjusted to 7.0 using 1 N NaOH) plates at 22°C for 15 days under a 12-h light/dark cycle. Similarly, *B. juncea* var. Varuna was grown at 25°C with a photoperiod of 10 h light/14 h dark and used for the infection assays.

**Disease inoculation assays and sample collection.** Briefly, 15-day-old PDA plates were used to prepare spore suspension for drop inoculations. Multiple 15-$\mu$L droplets of the spore suspension (concentration of $3 \times 10^3$ to $5 \times 10^3$ conidia/mL) were placed on the leaves of 5- to 6-week-old *B. juncea* plants to maximally cover the surface area. The leaves were then maintained at >90% relative humidity to enable infection. Leaves

were collected 2 and 4 days post inoculation (2 and 4 dpi). Fifteen-day-old *A. brassicae* mycelium growing on PDA was also harvested, which contained a mixed population of growing mycelia and spores (*in vitro* sample). A total of three independent experiments were carried out with each experiment containing 5 to 6 leaves from three different plants for each time point.

**RNA extraction, library preparation, and sequencing.** Total RNA was extracted from 5 to 6 leaves collected from three individual plants in each experiment using the MagMAX-96 total RNA isolation kit according to the manufacturer's recommendation (Invitrogen, Thermo Fisher Scientific, Waltham, MA, USA). Total RNA (500 ng) was used to enrich the mRNA using NEBNext poly(A) mRNA magnetic isolation module (New England Biolabs, Ipswich, MA, USA) by following the manufacturer's protocol. Libraries were prepared from the enriched mRNAs using the NEBNext Ultra II RNA library prep kit for Illumina (New England Biolabs, Ipswich, MA, USA). The library concentration was determined in a Qubit3 fluorometer using the Qubit double-stranded DNA (dsDNA) high-sensitivity (HS) assay kit (Thermo Fisher Scientific, Waltham, MA, USA). The library quality was assessed using an Agilent D5000 ScreenTape system in a 4150 TapeStation system. All the libraries were sequenced on an Illumina NovaSeq platform (Clevergene Pvt. Ltd., Bengaluru, India).

**Reannotation of the reference genome with *in planta* transcript evidence.** The transcriptome sequencing (RNA-Seq) reads from nine libraries (three biological replicates per condition) were processed to remove adaptor sequences, empty reads, and low-quality sequences with a Phred score of <30 and reads of <36 bp. The processed reads were stored in FASTQ format. The processed reads were mapped to the reference genome obtained from a previous report (16, 22) using HISAT2 (45). The resulting bam files were then used as inputs to stringtie2 (46) to assemble the alignments into transcripts. The transcripts were then used as hints in the funannotate eukaryotic genome prediction and annotation pipeline to predict genes (47). The new gene predictions were then merged with the earlier gene predictions (22) to obtain a revised annotation of the *A. brassicae* genome, which is referred to as V 2.0.

**Transcript quantification and differential gene expression analysis.** The processed reads were mapped to the reference genome (16) using HISAT2, and the resulting bam files were used to assemble and quantify the transcripts using stringtie2 default parameters. The FPKM (fragments per kilobase per million) values from stringtie gff output were converted to raw count values using the prepDE.py script supplied along with the stringtie2 software. Differential gene expression (DGE) analysis was conducted using the DESeq2 package (48). The counts data obtained from stringtie2 were used as the input for DESeq2. The raw counts were transformed using the variance stabilizing transformation (vst) implemented in the DESeq2 package. The transformed counts were used to carry out a principal-component analysis (PCA) to determine the relatedness of the biological replicates using the DESeq2 package in R with default parameters. A false-discovery rate (FDR) cutoff of 0.05 was applied to account for multiple testing corrections. Genes with a $\geq$2-fold (absolute value of $\log_2$ fold change |LFC| of $\geq$1) change in expression level and an FDR-adjusted *P* value of <0.05 were considered "differentially expressed."

**Functional classification and enrichment analyses of DEGs.** Gene ontology (GO) enrichment analysis was performed using the GOstats package (49). GO terms assigned to the total gene list of *A. brassicae* were used as the background list for enrichment analysis. The hypergeometric test implemented in GOstats was used to identify significantly enriched GO categories. A GO category was considered significantly enriched only when the *P* value for that category was <0.05.

**Weighted gene coexpression network analysis of DEGs.** Gene coexpression networks were constructed using the WGCNA package (50) in R. The vst-transformed expression data for the DEGs were used for coexpression analysis. A soft thresholding power of 16 was determined using the pickSoftThreshold function based on the scale-free topology model fit ($r^2 > 0.9$). The blockwiseModules function was used to obtain coexpression modules, and closely related modules were merged at a cutHeight of 0.20. Pearson's correlation coefficient was calculated to correlate the modules with the sample traits. GO term enrichment analysis was carried out using a standard hypergeometric test as implemented in the GOstats package. Cytoscape v3.8.2 (51, 52) was used to visualize the coexpression networks.

**Data availability.** The transcriptome data set of all the samples has been deposited in the NCBI BioProject database (BioProject number PRJNA860050).

## SUPPLEMENTAL MATERIAL

Supplemental material is available online only.
**SUPPLEMENTAL FILE 1**, PDF file, 1.3 MB.
**SUPPLEMENTAL FILE 2**, XLSX file, 0.5 MB.

## ACKNOWLEDGMENTS

This research was supported by grants from the Department of Science and Technology through the DST-INSPIRE Faculty program to S.R. and a Core Research Grant from the Science and Engineering Research Board (SERB grant no. CRG/2020/001731).

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
