## [Reviewer comments · Microbiology Spectrum]

Microbiology Spectrum

Transcriptome analysis of the necrotrophic pathogen *Alternaria brassicae* reveals insights into its pathogenesis in *Brassica juncea*

Sivasubramanian Rajarammohan

Corresponding Author(s): Sivasubramanian Rajarammohan, National Agri-Food Biotechnology Institute

Review Timeline:

Submission Date:	July 29, 2022
Editorial Decision:	September 9, 2022
Revision Received:	September 29, 2022
Editorial Decision:	December 6, 2022
Revision Received:	February 3, 2023
Accepted:	February 11, 2023

Editor: Giuseppe Ianiri

Reviewer(s): The reviewers have opted to remain anonymous.

Transaction Report:

DOI: <https://doi.org/10.1128/spectrum.02939-22>

Dr. Sivasubramanian Rajarammohan
National Agri-Food Biotechnology Institute
MOHALI
India

Re: Spectrum02939-22 (Transcriptome analysis of the necrotrophic pathogen *Alternaria brassicae* reveals a biphasic mode of pathogenesis in *Brassica juncea*)

Dear Dr. Sivasubramanian Rajarammohan:

I have received the reviews of your manuscript entitled "Transcriptome analysis of the necrotrophic pathogen *Alternaria brassicae* reveals a biphasic mode of pathogenesis in *Brassica juncea*", and I regret to inform you that we will not be able to publish it in Spectrum. Your submission was read by reviewers with expertise in the area addressed in your study and it was the consensus view of these reviewers that your paper did not meet the standards necessary for publication. Copies of the reviewers' comments are attached for your consideration.

I am sorry to convey a negative decision on this occasion, but I hope that the enclosed reviews are useful. Please note, rejections from Microbiology Spectrum are final and your manuscript will not be considered by other ASM journals. We wish you well in publishing this report in another journal and hope that you will consider Spectrum in the future.

Sincerely,

Giuseppe Ianiri
Editor, Microbiology Spectrum

Reviewer comments:

Reviewer #1 (Comments for the Author):

This paper has too many flaws and is not suitable for publication in Microbiology Spectrum. The method section is inadequate. The data (PRJNA860050) is not publicly available. For instance how many reads were obtained from how many libraries. The experimental design, replicates etc. are not discussed. PCA was carried out with what software and what parameter was not discussed. The total number of genes and the genes having expression in control vs. infected samples are not clearly discussed. Initially I was documenting what is wrong with the paper, later I lost count and stopped writing.

Reviewer #2 (Comments for the Author):

In this study, the author investigated the differentially expressed genes during growth and infection in the *Alternaria* blight or leaf spot pathogen *Alternaria brassicae*. The author listed the biological processes after GO and WGCNA analysis. Improving the understanding of *Alternaria brassicae* in planta expression is an important research area. However, a substantial amount of functional analysis should be accomplished. There is a general lack of description in approaches used and there are minimal details provided in the Methods or Figure Legends.

In this study, the author investigated the differentially expressed genes during growth and infection in the *Alternaria* blight or leaf spot pathogen *Alternaria brassicae*. The author listed the biological processes after GO and WGCNA analysis. Improving the understanding of the in planta expression is an important research area. However, a substantial amount of functional analysis should be accomplished. There is a general lack of description in approaches used and there are minimal details provided in the Methods or Figure Legends. Some specific issues are outlined below:

1 The author performed transcriptome sequencing at 2 or 4 days post inoculation, designated as the early and late time points of infection. But the author did not provide any evidence to prove the sampling time points represented as the early and late infection. In addition, the sampling should be described. The mycelia from PDA plate included the aerial and feeding hyphae, which introduced the variety of differentially expressed genes.

2 When analyzed the upregulated genes at 2 and 4 dpi, the author did not mention the common genes up- or down- regulated at 2 and 4 dpi. This should be addressed here and as the in planta expression needed.

3 The author state that "In contrast, three genes, ABRSC02.163, ABRSC01.1330, and ABRSC09.497, were significantly downregulated at 2 dpi but were upregulated at 4 dpi" However, it doesn't appear that this data has been indicated.

4 For GO and WGCNA analysis, the author only list the gene modules, but not described their differences at different stages and with different expression level. What are their expression patterns during infection? What are those genes function in infection?

5 For the candidate genes, is the GO and WGCNA analysis important to identify those genes? What is the relationship of your network analysis with the biological process and those pathogenesis genes? The list of these genes has been described in many papers, and there were nothing novel. What are the common or specific pathways /module in the necrotrophic pathogens?

Point-by-point rebuttal of reviewer comments –

Reviewer 1:

Comment: The data (PRJNA860050) is not publicly available.

Response: The data has been submitted under the above said BioProject ID and as is common in many studies, the data will be released upon publication. In case the reviewer needs access to the raw data, NCBI provides a separate reviewer link for the same (Reviewer link to PRJNA860050:

<https://dataview.ncbi.nlm.nih.gov/object/PRJNA860050?reviewer=6flkf2ujumi75b74llqms14923>).

Comment: For instance how many reads were obtained from how many libraries.

Response: The total reads obtained have been mentioned in Page 4, Line 101. The details of reads from each library is mentioned in Table S1 (Page 4, Line 102).

Comment: The experimental design, replicates etc. are not discussed.

Response: The experimental design and replicates are described in Figure 1. The no. of replicates (3) is also mentioned in Figure 1. Additionally, the sample collection and replicates are mentioned in the Methods section on Page 13, Lines 353-356.

Comment: PCA was carried out with what software and what parameter was not discussed.

Response: PCA was performed as part of the DESeq2 package to identify differentially expressed genes and has been mentioned in the Methods section in Page 14, Lines 381-382.

Comment: The total number of genes and the genes having expression in control vs. infected samples are not clearly discussed.

Response: The criteria for “expressing” genes and the total number of genes expressed (11,124) is given in Page 4, Line 106-108. The number of genes expressed in infected or in planta samples (8,603) is given on Page 5, Line 109.

Reviewer 2:

Comment: The author performed transcriptome sequencing at 2 or 4 days post inoculation, designated as the early and late time points of infection. But the author did not provide any evidence to prove the sampling time points represented as the early and late infection. In addition, the sampling should be described. The mycelia from PDA plate included the aerial and feeding hyphae, which introduced the variety of differentially expressed genes.

Response: The logic for choosing 2 and 4 dpi as early and late time points has been explained in Page 4, Line 96-99 and further discussed in Page 9, Line 245-249. Specifically, “We chose the 2 and 4 days post inoculation (dpi) as the early and late time points of infection since previous studies have shown that *A. brassicae* spores maximally penetrate the host tissue at 2 dpi and visible symptoms are clearly visible at 4 dpi (17).” The sampling procedure has been described in Figure 1 and in the Methods section on Page 13 Lines 353-356.

Comment: When analyzed the upregulated genes at 2 and 4 dpi, the author did not mention the common genes up- or down-regulated at 2 and 4 dpi. This should be addressed here and as the in planta expression needed.

Response: The common genes up- or down-regulated at 2 and 4 dpi have been depicted in Figure 3A as a Venn diagram and the top up-regulated genes in 2 and 4 dpi are depicted in Figure 3B as a heatmap and also in Table S2 and Table S3. Additionally, the common and unique genes are explained in Page 5, Line 120-127.

Comment: The author state that "In contrast, three genes, ABRSC02.163, ABRSC01.1330, and ABRSC09.497, were significantly downregulated at 2 dpi but were upregulated at 4 dpi" However, it doesn't appear that this data has been indicated.

Response: The data pertaining to the above statement is indicated in Supplementary Table S2 and referenced at the end of the statement on Page 5, Lines 128 and 130.

Comment: For GO and WGCNA analysis, the author only lists the gene modules, but not described their differences at different stages and with different expression level. What are their expression patterns during infection? What are those genes function in infection?

Response: The WGCNA was performed only for the DEGs. The modules have been correlated to the time points of infection and have been listed in the same order. The modules M1 and M2 were positively correlated to the growth on PDA plates, whereas M6, M7, and M8 were positively correlated to the early time point of infection. The M9 module was positively correlated to the late time point of infection. This has been explained in the results section in Page 6, Line 156-166.

The GO analysis indicates the functions of the genes that are overrepresented in the respective modules. The specific biological processes of the genes in specific modules have also been described in Page 6 and 7, Line 167-179. The details of the GO enrichment analysis of the modules is also given in Figure 5 and Table S8. Additionally, the model described in Figure 7 lists the specific modules and GO processes that are active at the different stages of infection.

Comment: For the candidate genes, is the GO and WGCNA analysis important to identify those genes? What is the relationship of your network analysis with the biological process and those pathogenesis genes? The list of these genes has been described in many papers, and there was nothing novel. What are the common or specific pathways /module in the necrotrophic pathogens?

Response: The GO and WGCNA analysis is independent of the list of candidate genes discussed. The candidate genes identified were classified into CAZymes, Effectors, and secondary metabolites. Although some of these genes may have been described in other papers, the functional expression of these genes in *A. brassicae* is shown here for the first time. Additionally, many of the effectors described in the study have no known domain or function and have not been reported elsewhere. Effectors of necrotrophic pathogens are not well described and the current study provides a foundation for further functional characterisation of these effectors. In case of the secondary metabolites, Destruxin B as well HC-toxin gene clusters are specifically present in *A. brassicae* and not in other infective *Alternaria* species. The current study thus provides proof of these secondary metabolite clusters to be activated during infection and may therefore encourage further studies on these important secondary metabolites.

December 6, 2022

Dr. Sivasubramanian Rajarammohan
National Agri-Food Biotechnology Institute
MOHALI
India

Re: Spectrum02939-22R1-A (Transcriptome analysis of the necrotrophic pathogen *Alternaria brassicae* reveals a biphasic mode of pathogenesis in *Brassica juncea*)

Dear Dr. Sivasubramanian Rajarammohan:

Three other experts reviewed your manuscript. All of them raised the issue that more time points should have been included to make the paper more significant, but in Microbiology Spectrum the impact of a manuscript is not the main point of evaluation. Reviewer 2 raised a serious question about the title of your manuscript, because it seems that this biphasic mode of infection is not well highlighted in the text. I agree with him, and strongly suggest to change the title of your manuscript. Moreover, all reviewers together raised a number of comments that you can find in the email below, and I can accept your manuscript only if an exhaustive answer will be provided to all their comments. Failing to address some comments will likely result in manuscript rejection, without the possibility of further appealing.

Link Not Available

Sincerely,

Giuseppe Ianiri

Journals Department
Reviewer comments:

Reviewer #3 (Public repository details (Required)):

The authors have produced an RNA-Seq dataset. They state it is uploaded to NCBI and provide an accession number, however it is not yet public so I am unable to verify that it will be available upon publication. The verification link provided in the previous response to reviewers does show files awaiting release, so on balance I think the author will release them, though I would prefer

they were made public on submission rather than upon acceptance, particularly as a preprint has been submitted.

Reviewer #3 (Comments for the Author):

The author has produced a RNA-Seq dataset of an infection time-course of *Alternaria brassicae* infecting *Brassica juncea*. The author has followed this with some initial analyses of the changes in expression and some basic functional annotations. Whilst this is an understudied pathosystem and I would follow similar initial exploratory analyses, I do think there are several improvements that can be made to the manuscript.

Major:

- 1) The author should explain which parameters were used for STAR as it is highly customisable. I would also like to know if a one or two pass method was used. Two-pass methods have previously been shown to improve the quantification accuracy of different splice variants of a gene.
- 2) I would like an explanation of why featurecounts was used. This is older software that simply pulls the number of reads aligning to a feature, whereas newer tools like stringtie feature newer algorithms that can further improve accuracy and suggest novel transcripts that could have been missed during annotation.
- 3) The author cites a note on the genome assembly (reference 16) of *A. brassicae* as being the source of the assembly and annotations, yet this note only describes the assembly. The author should provide the source of the annotations or describe how they were generated if this is novel work. - I think this is covered by reference 22, but it is not cited at line 380.
- 4) The author should clarify in the WGCNA methodology whether a false discovery rate was again used for multi-test correction when assessing over-representation.

Minor:

- 1) At line 70, the author has already explained that knowledge of infection is limited and yet does so again in the next sentence. This hurts the readability of the manuscript and could be rephrased.
- 2) The author frequently uses plural pronouns eg. we despite this being a single author manuscript. If others conducted the analyses they should receive appropriate credit. If not, this should be rephrased to avoid confusion to a reader.
- 3) I would like the author to explain in more detail how the PCA was performed and whether any parameters were passed to the analysis.
- 4) The author should reword mentions of effectors, CAZyS and Toxins to make clear these are previously identified rather than new in this paper.

Reviewer #4 (Public repository details (Required)):

Data seems to have been deposited in NCBI

Reviewer #4 (Comments for the Author):

The manuscript entitled "Transcriptome analysis of the necrotrophic pathogen *Alternaria brassicae* reveals a biphasic mode of pathogenesis in *Brassica juncea*" aim to provides novel insights into the transcriptional plasticity of a necrotrophic pathogen during infection of its host.

To be honest I am a bit confused as the title clearly state: "biphasic mode of pathogenesis", therefore I was assuming that the main novelty of this work is the clear description of a new pathogenicity strategy in *Alternaria brassicae*. However, nothing is reported about a biphasic mode of pathogenesis in the abstract... moreover more I was reading the manuscript more I was consolidating my doubts about a biphasic mode of pathogenesis.

The author used a transcriptomic approach using 2 time points and comparing the expression profiles of *Alternaria brassicae* to the fungus growing on PDA.

To describe a "biphasic mode of pathogenesis" I would have expected more timepoints.

It's quite funny that the author found a biphasic mode using two time points... I am wondering if he could have found a triphasic mode using three time points or a fourphasic mode using four time points.

Stating that a pathogen has a biphasic mode means a clear switch between two different phases. An example of a biphasic lifestyles is normally used for pathogens such as *Colletotrichum* and *Magnaporthe*, in those systems the fungus differentiates

distinctive structures and activate different genes during different stages (However also in these cases the lifestyle is much more complex).

That does not mean that necrotrophic pathogens have a unique expression profile during the interaction with the host, still necrotrophic pathogens do not differentiate distinctive structures and no clear switch can be detected. Assuming that necrotrophy is a simple "kill and eat" process with no molecular cross-talk, gene modulation, and complexity is quite simplistic and old fashion.

The classification of different trophic lifestyles (like any other biological classification) is a strategy that does not reflect the complexity of biology. If the author wants to have an alternative approach to classification, maybe he/she should look at the different shades that biology has... rather than increase the number of classes.

Having said that there are also technical issues in the manuscript that should be considered. A few examples are:

The author cites their work for the identification of selected gene families like the CAZY. I checked the published work and it state "Genes were then annotated using BLAST (version 2.7.1+) against UniProt, SWISS-PROT, CAZY, MEROPS, and PHI-BASE." Not considering that nothing is stated about how the BLAST was performed (score, e-value or similarity cutoffs) I would not trust such an approach for the characterization of CAZY and peptidases as the similarity may be misleading and usually a more complex approach (sequence alignments, hidden markov models (HMMs) and phylogenies of the signature domain regions). The characterization of both those gene families is much more complex.

The author analyzes candidate effectors, CAZY and SM clusters... why not all SM associated backbone genes and no peptidases were analyzed? How did he/she select specific gene families and not others?

The manuscript is also quite hard to read and follow, an example is provided by the numerous time the author state "during infection" in the abstract or the structure of the images.

Reviewer #5 (Public repository details (Required)):

The author states that the transcriptome dataset has been deposited in the NCBI BioProject database.

Reviewer #5 (Comments for the Author):

In this publication, the author performed transcriptomic analysis of *Alternaria brassicae* on *Brassica juncea* during two infection time points. These data should fill a gap in the field on molecular mechanisms of pathogenicity of *A. brassicae*. Previous reports have focused on secondary metabolite toxins and the host response to *A. brassicae*.

Major comments:

- 1, The author responded to the comments of the reviewers by including more information in the Methods and Results section. However, the author did not clearly state if there was a mixed population of *A. brassicae* on the 15-day-old PDA plate at the time of sampling.
- 2, Regarding the timing of the infection and collecting the samples, this manuscript would benefit from an earlier time point during infection as a control, instead of an artificial media at the negative control. One could envision that collecting the sample directly after inoculation and prior to penetration at day 2, would also make the populations more consistent.
- 3, In this discussion, the author states that "approximately 59% of the putative effectors were expressed at least at one of the time points. This indicates that proteinaceous effectors are vital during the pathogenesis of *A. brassicae*." A comparison to percentage of effectors in other necrotrophic pathogens would be helpful, as would some functional assays that provide evidence of the role of effector proteins in *A. brassicae* pathogenicity. However, it is understandable that the latter is outside of the scope of this paper.

Staff Comments:

Preparing Revision Guidelines

Please return the manuscript within 60 days; if you cannot complete the modification within this time period, please contact me. If you do not wish to modify the manuscript and prefer to submit it to another journal, please notify me of your decision immediately so that the manuscript may be formally withdrawn from consideration by Microbiology Spectrum.

In this publication, the author performed transcriptomic analysis of *Alternaria brassicae* on *Brassica juncea* during two infection time points. These data should fill a gap in the field on molecular mechanisms of pathogenicity of *A. brassicae*. Previous reports have focused on secondary metabolite toxins and the host response to *A. brassicae*.

Major comments:

1, The author responded to the comments of the reviewers by including more information in the Methods and Results section. However, the author did not clearly state if there was a mixed population of *A. brassicae* on the 15-day-old PDA plate at the time of sampling.

2, Regarding the timing of the infection and collecting the samples, this manuscript would benefit from an earlier time point during infection as a control, instead of an artificial media at the negative control. One could envision that collecting the sample directly after inoculation and prior to penetration at day 2, would also make the populations more consistent.

3, In this discussion, the author states that “approximately 59% of the putative effectors were expressed at least at one of the time points. This indicates that proteinaceous effectors are vital during the pathogenesis of *A. brassicae*.” A comparison to percentage of effectors in other necrotrophic pathogens would be helpful, as would some functional assays that provide evidence of the role of effector proteins in *A. brassicae* pathogenicity. However, it is understandable that the latter is outside of the scope of this paper.

In this publication, the author performed transcriptomic analysis of *Alternaria brassicae* on *Brassica juncea* during two infection time points. These data should fill a gap in the field on molecular mechanisms of pathogenicity of *A. brassicae*. Previous reports have focused on secondary metabolite toxins and the host response to *A. brassicae*.

Major comments:

1, The author responded to the comments of the reviewers by including more information in the Methods and Results section. However, the author did not clearly state if there was a mixed population of *A. brassicae* on the 15-day-old PDA plate at the time of sampling.

2, Regarding the timing of the infection and collecting the samples, this manuscript would benefit from an earlier time point during infection as a control, instead of an artificial media at the negative control. One could envision that collecting the sample directly after inoculation and prior to penetration at day 2, would also make the populations more consistent.

3, In this discussion, the author states that “approximately 59% of the putative effectors were expressed at least at one of the time points. This indicates that proteinaceous effectors are vital during the pathogenesis of *A. brassicae*.” A comparison to percentage of effectors in other necrotrophic pathogens would be helpful, as would some functional assays that provide evidence of the role of effector proteins in *A. brassicae* pathogenicity. However, it is understandable that the latter is outside of the scope of this paper.

Response to reviewers

Reviewer #3 (Public repository details (Required)):

Comment: The authors have produced an RNA-Seq dataset. They state it is uploaded to NCBI and provide an accession number, however it is not yet public so I am unable to verify that it will be available upon publication. The verification link provided in the previous response to reviewers does show files awaiting release, so on balance I think the author will release them, though I would prefer they were made public on submission rather than upon acceptance, particularly as a preprint has been submitted.

Response: As per the reviewer's suggestion, the dataset uploaded to NCBI has been made public now.

Reviewer #3 (Comments for the Author):

The author has produced a RNA-Seq dataset of an infection time-course of *Alternaria brassicae* infecting *Brassica juncea*. The author has followed this with some initial analyses of the changes in expression and some basic functional annotations. Whilst this is an understudied pathosystem and I would follow similar initial exploratory analyses, I do think there are several improvements that can be made to the manuscript.

Major comments -

Comment: The author should explain which parameters were used for STAR as it is highly customisable. I would also like to know if a one or two pass method was used. Two-pass methods have previously been shown to improve the quantification accuracy of different splice variants of a gene.

Response: I had initially compared the one and two pass methods in STAR. However, I did not find any major improvement in the number of reads that mapped to the transcripts. The marginal increase in the reads that mapped using the two-pass method resulted in the increase in the transcript count for the highly abundant transcripts only and did not map to any transcript that was not assigned any reads earlier. However, in response to the reviewer suggestion in point no. 2 I have carried out the analysis using HISAT2 + stringtie2 and this has been explained in the response to the below comment.

Comment: I would like an explanation of why featurecounts was used. This is older software that simply pulls the number of reads aligning to a feature, whereas newer tools like stringtie feature newer algorithms that can further improve accuracy and suggest novel transcripts that could have been missed during annotation.

Response: I thank the reviewer for this suggestion. Even though featurecounts is an older software it is still being extensively used to generate raw counts from RNASeq data and was therefore used in this study. However, as per the reviewer's suggestion, the analysis was carried out using Stringtie2, which resulted in the discovery of transcripts that were not predicted/annotated in the previous study. Therefore, a re-annotation of the genome was

carried out using this transcriptome dataset to obtain a revised version of the genome annotation of *A. brassicae* (V 2.0). The revised version of the annotations was used for all further analyses. Further, the transcript quantification was also carried out using stringtie2 and the transcript counts obtained from stringtie2 were used in DESeq2 for differential gene expression analysis. Although the new analysis revealed more DEGs, the functional classification, and GO enrichment did not change drastically.

Comment: The author cites a note on the genome assembly (reference 16) of *A. brassicae* as being the source of the assembly and annotations, yet this note only describes the assembly. The author should provide the source of the annotations or describe how they were generated if this is novel work. - I think this is covered by reference 22, but it is not cited at line 380.

Response: I apologise for this omission. This has been rectified in the revised manuscript.

Comment: The author should clarify in the WGCNA methodology whether a false discovery rate was again used for multi-test correction when assessing over-representation.

Response: The genes identified in each module in the WGCNA methodology was subject to GO enrichment analysis using a standard hypergeometric test as implemented in the GOSTats package. This information has now been included in the revised manuscript. One of the underlying assumptions of the hypergeometric is that the individual terms/genes being chosen are independent. Unfortunately, this is not true of genes or GO terms. Both the genes and GO terms are interdependent (expression of one gene affecting other(s) and GO terms are set up as a directed acyclic graph). Due to this interdependency, carrying out multiple testing on GO datasets are conceptually challenging. The use of FDR is useful but it does not solve the issue of dependency. Additionally, the sampling of genes used in the analysis has already been subjected to statistical testing and multiple correction (only the DEGs and not all the genes were used for network construction using WGCNA) and the correlation of genes to their functional attributes (as determined by GO terms) is deterministic. I have also only attributed functional categories that are represented by multiple genes (in the same module) in the manuscript. Therefore, FDR correction was not applied when assessing the over-representation.

Minor comments -

Comment: At line 70, the author has already explained that knowledge of infection is limited and yet does so again in the next sentence. This hurts the readability of the manuscript and could be rephrased.

Response: I had referred to two different pathogens at Line 68-71 viz. *A. brassicicola* and *A. brassicae*. However, as per the reviewer's suggestion, I have rephrased the sentences for better readability in the revised manuscript.

Comment: The author frequently uses plural pronouns eg. we despite this being a single author manuscript. If others conducted the analyses they should receive appropriate credit. If not, this should be rephrased to avoid confusion to a reader.

Response: I apologise for this error. All the instances of “we” has been changed to avoid confusion to the reader.

Comment: I would like the author to explain in more detail how the PCA was performed and whether any parameters were passed to the analysis.

Response: The PCA was performed as is implemented in the DESeq2 package. The raw counts were transformed using the variance stabilising transformation and the transformed values were used to generate the PCA plot using default parameters. This has been now included in the revised manuscript.

Comment: The author should reword mentions of effectors, CAZys and Toxins to make clear these are previously identified rather than new in this paper.

Response: I thank the reviewer for pointing this out. This has been rectified in the revised manuscript.

Reviewer #4 (Public repository details (Required)):

Data seems to have been deposited in NCBI

Reviewer #4 (Comments for the Author):

The manuscript entitled "Transcriptome analysis of the necrotrophic pathogen *Alternaria brassicae* reveals a biphasic mode of pathogenesis in *Brassica juncea*" aim to provides novel insights into the transcriptional plasticity of a necrotrophic pathogen during infection of its host.

Comment: To be honest I am a bit confused as the title clearly state: "biphasic mode of pathogenesis", therefore I was assuming that the main novelty of this work is the clear description of a new pathogenicity strategy in *Alternaria brassicae*. However, nothing is reported about a biphasic mode of pathogenesis in the abstract... moreover more I was reading the manuscript more I was consolidating my doubts about a biphasic mode of pathogenesis. The author used a transcriptomic approach using 2 time points and comparing the expression profiles of *Alternaria brassicae* to the fungus growing on PDA. To describe a "biphasic mode of pathogenesis" I would have expected more timepoints. It's quite funny that the author found a biphasic mode using two time points... I am wondering if he could have found a triphasic mode using three time points or a fourphasic mode using four time points. Stating that a pathogen has a biphasic mode means a clear switch between two different phases. An example of a biphasic lifestyles is normally used for pathogens such as *Colletotrichum* and *Magnaporthe*, in those systems the fungus differentiates distinctive structures and activate different genes during different stages (However also in these cases the lifestyle is much more complex). That does not mean that necrotrophic pathogens have a unique expression profile during the interaction with the host, still necrotrophic pathogens do not differentiate distinctive structures and no clear switch can be detected. Assuming that necrotrophy is a simple "kill and eat" process with no molecular cross-talk, gene modulation, and complexity is

quite simplistic and old fashion. The classification of different trophic lifestyles (like any other biological classification) is a strategy that does not reflect the complexity of biology. If the author wants to have an alternative approach to classification, maybe he/she should look at the different shades that biology has... rather than increase the number of classes.

Response: I thank the reviewer for his/her suggestions. I agree that more time points are required to claim a “biphasic” mode of pathogenesis and have hence changed the title of the manuscript and removed all the instances of a biphasic mode in the revised manuscript. I proposed a “biphasic” mode of pathogenesis for *A. brassicae* not only from the transcriptome data generated in this study but also previous detailed histopathological studies that clearly showed that the pathogen had a short “biotrophic” phase where it grew within the host (intercellularly) without causing cell death. Apart from *A. brassicae*, this has also been reported for the broad host-range necrotrophic pathogen *Sclerotinia sclerotiorum* by other research groups. Although, *A. brassicae* does not develop very distinct structures during pathogenesis, the histopathological studies did show the formation of penetration pegs or appresoria-like structures that may perhaps indicate the development stage during which the “switch” may occur. However, this is speculative with the current information and would warrant further fine-scale transcriptome studies. I also agree with the reviewer that any type of classification does not fully reflect the complexity of biology. My intention was to draw attention to the aforementioned complexities during interactions of *A. brassicae* with its host and move away from the assumption of “kill & eat” process that the reviewer rightly calls as simplistic and old fashioned.

Comment: The author cites their work for the identification of selected gene families like the CAZy. I checked the published work and it state "Genes were then annotated using BLAST (version 2.7.1+) against UniProt, SWISS-PROT, CAZy, MEROPS, and PHI-BASE." Not considering that nothing is stated about how the BLAST was performed (score, e-value or similarity cutoffs) I would not trust such an approach for the characterization of CAZy and peptidases as the similarity may be misleading and usually a more complex approach (sequence alignments, hidden markov models (HMMs) and phylogenies of the signature domain regions). The characterization of both those gene families is much more complex.

Response: The CAZymes were identified by searching against the CAZy database maintained at www.cazy.org. The CAZymes and peptidases (from MEROPS) were identified using BLAST with a cutoff of percent identity > 50% and a query coverage > 50% to rule out false positive matches. Additionally, the CAZymes were also annotated by using the dbCAN web server, which uses a signature domain-based annotation. The signature domain for every CAZyme family is derived based on the CDD (conserved domain database) search and literature curation. Additionally, dbCAN also contains HMMs of the signature domains, which are used to annotate a given set of proteins. The results from the BLAST search against CAZy database and the annotations from the dbCAN server were in concordance with a few extra CAZymes that were identified in the dbCAN approach. This set of CAZymes were only investigated in the current study. A similar approach was also used for the peptidases, wherein the BLAST search was carried out using only the signature domains of the peptidases as the reference.

Comment: The author analyzes candidate effectors, CAZY and SM clusters... why not all SM associated backbone genes and no peptidases were analyzed? How did he/she select specific gene families and not others?

Response: The SM-associated backbone genes were analysed and their details are given in Supplementary Table S11. Specific genes and gene families that were already reported to have a role in pathogenesis were selected and discussed. For example, pectate lyases have been reported to have a role in pathogenesis of related pathogen *A. brassicicola*. Even though specific gene families have been discussed, the data on expression of all CAZymes, effectors, and SM clusters have been recorded in the Supplementary Tables S9-S12. Peptidases were not analysed earlier, but as per the reviewer's suggestion this has been carried out and is included in the revised manuscript (Line nos. 237-246; Table S12).

Comment: The manuscript is also quite hard to read and follow, an example is provided by the numerous time the author state "during infection" in the abstract or the structure of the images.

Response: The manuscript has been revised for structural changes in the language and repetitions have been avoided wherever possible.

Reviewer #5 (Public repository details (Required)):

The author states that the transcriptome dataset has been deposited in the NCBI BioProject database.

Reviewer #5 (Comments for the Author):

In this publication, the author performed transcriptomic analysis of *Alternaria brassicae* on *Brassica juncea* during two infection time points. These data should fill a gap in the field on molecular mechanisms of pathogenicity of *A. brassicae*. Previous reports have focused on secondary metabolite toxins and the host response to *A. brassicae*.

Major comments -

Comment: The author responded to the comments of the reviewers by including more information in the Methods and Results section. However, the author did not clearly state if there was a mixed population of *A. brassicae* on the 15-day-old PDA plate at the time of sampling.

Response: *A. brassicae* collected from 15-day-old PDA plate had a mixed population of growing mycelia and spores at the time of sampling. This information has now been included in the revised manuscript.

Comment: Regarding the timing of the infection and collecting the samples, this manuscript would benefit from an earlier time point during infection as a control, instead of an artificial media at the negative control. One could envision that collecting the sample directly after

inoculation and prior to penetration at day 2, would also make the populations more consistent.

Response: I thank the reviewer for this suggestion. However, there are two major hindrances in collecting an earlier time point. The *A. brassicae* spores start to adhere strongly to the surface of the host tissue only after 24 hours of inoculation and are maximally adhered and start to penetrate the tissue at 2 dpi. I did collect samples from 24 hours and 36 hours post inoculation, however, the representation of fungal RNA from these samples was negligible and therefore could not be used for further analysis. Enrichment of fungal RNA from such samples has proved to be challenging and we are exploring other methodological improvements to sample and assay the gene expression changes in the early time points of infection.

Comment: In this discussion, the author states that "approximately 59% of the putative effectors were expressed at least at one of the time points. This indicates that proteinaceous effectors are vital during the pathogenesis of *A. brassicae*." A comparison to percentage of effectors in other necrotrophic pathogens would be helpful, as would some functional assays that provide evidence of the role of effector proteins in *A. brassicae* pathogenicity. However, it is understandable that the latter is outside of the scope of this paper.

Response: I thank the reviewer for this suggestion. The comparison of effectors (expressing during infection) from other necrotrophic pathogens has now been added to the discussion. As the reviewer pointed out the evidence of role of effectors in *A. brassicae* is out of scope of this paper but will surely be followed up in subsequent studies from the lab.

February 11, 2023

Dr. Sivasubramanian Rajarammohan
National Agri-Food Biotechnology Institute
MOHALI
India

Re: Spectrum02939-22R2 (Transcriptome analysis of the necrotrophic pathogen *Alternaria brassicae* reveals insights into its pathogenesis in *Brassica juncea*)

Dear Dr. Sivasubramanian Rajarammohan:

Thanks for modifying the manuscript according to the reviewer's comments. I think that now the manuscript is different and much better than the one you originally submitted and that was rejected. In this journal the process of peer review is fair and its only purpose is to improve the quality of the science that we publish and put out there for the scientific community.

Your manuscript has been accepted, and I am forwarding it to the ASM Journals Department for publication. You will be notified when your proofs are ready to be viewed.

Sincerely,

Giuseppe Ianiri
Editor, Microbiology Spectrum
